# Towards Cross-Tokenizer Distillation: the Universal Logit Distillation Loss for LLMs

**Nicolas Boizard**  *nicolas.boizard@centralesupelec.fr*
*Diabolocom, Paris, France*
*MICS, CentraleSupélec, Paris-Saclay University, France*

**Kevin El Haddad**  *kevin.elhaddad@diabolocom.com*
*Diabolocom, Paris, France*
*ISIA Lab - University of Mons, Belgium*

**Céline Hudelot**  *celine.hudelot@centralesupelec.fr*
*MICS, CentraleSupélec, Paris-Saclay University, France*

**Pierre Colombo**  *pierre@equall.ai*
*Equall.ai*
*MICS, CentraleSupélec, Paris-Saclay University, France*

**Reviewed on OpenReview:** *https://openreview.net/forum?id=bwRxXiGO9A*

## Abstract

Deploying large language models (LLMs) with billions of parameters is often impractical in industrial settings due to constraints like cost, latency, and hardware limitations. Knowledge distillation (KD) provides a solution by compressing the knowledge from large, resource-intensive models into task-specific smaller ones. Various strategies exist, some relying on the text generated by the teacher model, optionally, leveraging its output logits to improve learning. However, these logit-based methods usually require the teacher and student models to share the same tokenizer, which limits their applicability across different model families. In this paper, we propose the Universal Logit Distillation (ULD) loss, which uses optimal transport theory to enable distillation across different architectures and tokenizers. Our results demonstrate that ULD loss effectively facilitates the distillation process, paving the way for a more widespread use of distillation.

## 1 Introduction

A significant trend in NLP involves the utilization of large language models (LLMs) such as LLama (Touvron et al., 2023a), Mistral (Jiang et al., 2023), Falcon (Almazrouei et al., 2023), GPT-NeoX (Black et al., 2022), or Mixtral (Jiang et al., 2024). While LLMs offer impressive performance (Bubeck et al., 2023), their deployment is often hampered by hardware availability, cost, and latency bottlenecks. Several strategies, such as efficient decoding (Leviathan et al., 2023; Ye et al., 2023), model recycling (Lester et al., 2022), and model size reduction (Dettmers et al., 2023; Ma et al., 2023), have been developed to streamline their use. Among these, knowledge distillation (KD) (Buciluundefined et al., 2006; Hinton et al., 2015) a widely adopted technique (Sanh et al., 2020; Jiao et al., 2020; Mohammadshahi et al., 2022; He et al., 2023; Raman et al., 2023; Dasgupta et al., 2023), transferring the capabilities of large, complex teacher models into more manageable and smaller student models, tailored for specific tasks. This approach is particularly valuable in contexts where the comprehensive knowledge contained within LLMs is not wholly necessary. This process aims to maintain the peak performance of general models on specific tasks while minimizing latency and memory usage.

Two approaches can be considered. The *"white box"* approach, where researchers propose loss functions that require access to the model architecture to compute similarities across layers, forcing adjustment for each situation. In contrast, the *"black box"* approach, indifferent to models latent spaces, relies solely on the output logit vectors from teacher and student. The *black-box* approach, due to its flexibility and generality, is easily implemented by practitioners through libraries or APIs, facilitating its adoption.

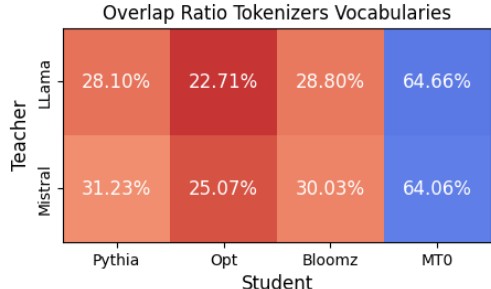

Over the past years, NLP researchers have extensively explored these distillation methods with teacher and student models sharing similar architecture, notably the BERT encoder (Sanh et al., 2020; Jiao et al., 2020; Sun et al., 2020). This desire to maintain similar architecture between teacher and student models by mirroring some of the teacher blocks, hidden sizes,

Figure 1: **Vocabulary overlap between teacher and student models** (Sec. 4.2), illustrating the challenge of cross-tokenizer distillation. (e.g., Bloomz tokenizer has 30.03% of Mistral's vocabulary).

or relying on the same tokenizer, came from the need to find similar information supports to apply distillation losses.

However, KD for generative models, those relying on encoder-decoder or decoder architectures, has received less attention due to the lack of common support between models. Although models of different sizes exist, they often diverge in architecture and tokenizer (Fig. 1), making logit distillation loss inapplicable. In fact, recent research predominantly focuses on synthetic data fine-tuning (He et al., 2022; Kramchaninova & Defauw, 2022; Ouyang et al., 2022), rather than refining logits loss in the black box approach. Thus far, the primary method for KD with decoder models is to use the text generated by the teacher model (He et al., 2023; Hsieh et al., 2023), and if possible, when students and teachers models belong to the same family, improving KD by employing output-logit distillation with Kullback–Leibler divergence (Mohammadshahi et al., 2022; Raman et al., 2023; Wang et al., 2020a). This raises the following research question:

*How can we build a general distillation loss that leverages logits capacity while staying within a black box framework that is easy to implement?*

**Contributions:** In this paper, we make the following contributions:

1. **A universal logit distillation loss**. We introduce a new loss, Universal Logit Distillation Loss (ULD loss), versatile to tokenizers and with minimal assumptions about the architectures of teacher and student models.

2. **Experimental Results**. We demonstrate the robust effectiveness of our loss function in transferring the capabilities of many teacher models to different smaller student models for a variety of specific tasks, including extractive question answering, generative question answering, and summarization.

3. **Contributing to future research**. We make our code[1], model weights, and generated datasets[2] openly available to facilitate future research, minimizing computational overhead and lowering entry barriers.

## 2 Problem Formulation & Related Work

### 2.1 Notations

We define $\Omega$ a model vocabulary set, $|\Omega|$ his size, and $\Omega^*$ signifies its Kleene closure[3]. Let $\mathcal{P}(\Omega)$ denote the set of probability distributions defined over the sample space $\Omega$. It is defined as $\mathcal{P}(\Omega) = \mathbf{p} \in [0,1]^\Omega$ with $\sum_{j=1}^{\Omega} p_j = 1$. Consider $\mathcal{D}$ as a non-empty set with independent and identically distributed samples

---

[1] https://github.com/Nicolas-BZRD/llm-recipes
[2] https://huggingface.co/Nicolas-BZRD
[3] The Kleene closure corresponds to sequences of arbitrary size written with words in $\Omega$. Formally: $\Omega^* = \bigcup_{i=0}^{\infty} \Omega^i$.

$(\mathbf{x}^i)_{i=1}^{N} \in \Omega^*$. Each $\mathbf{x}^i$ is a sequence of tokens, where $x_j^i \in \Omega$ represents the $j$th token of the $i$th sequence. The notation $\mathbf{x}_{<t}^i = (x_{t-1}^i, \ldots, x_0^i) \in \Omega^*$ denotes the prefix of length $t$. In this paper, we will denote by $\Omega^S$ the student vocabulary and by $\Omega^T$ the teacher's vocabulary.

**Remark.** *In general $\Omega^T \neq \Omega^S$ and $|\Omega^T| \neq |\Omega^S|$ but $\Omega^T \cap \Omega^S \neq \emptyset$.*

**Conditional textual generation:**  Conditional textual generation aims to model the probability distribution $\mathbf{p}_\star(\mathbf{x})$ over variable-length text sequence $\mathbf{x}$ by approximating $\mathbf{p}_{\boldsymbol{\theta}}(\mathbf{x})$ parameterized by $\theta \in \Theta$ to $\mathbf{p}_\star(\mathbf{x})$ for any $\mathbf{x}$. In this work, we assume the presence of a teacher $(f_{\boldsymbol{\theta_T}})$ and student $(f_{\boldsymbol{\theta_S}})$, two pre-trained conditional language models with a general form applicable to both $f_{\boldsymbol{\theta}} : \Omega^* \rightarrow \mathbb{R}^{|\Omega|}$, where the output is provided in unnormalized logit score. The function $f_{\boldsymbol{\theta}}$ parameterizes $\mathbf{p}_{\boldsymbol{\theta}}$ ($f_{\boldsymbol{\theta_T}}$ and $f_{\boldsymbol{\theta_S}}$ respectively parameterize $\mathbf{p}_{\boldsymbol{\theta_S}}$ and $\mathbf{q}_{\boldsymbol{\theta_T}}$), i.e., for any sentence $\mathbf{x}$, $\mathbf{p}_{\boldsymbol{\theta}}(\mathbf{x}) = \text{softmax}\left(\frac{f_{\boldsymbol{\theta}}(\mathbf{x})}{\tau}\right)$, where $\tau \in \mathbb{R}^+$ denotes the temperature set as default to 1. Given an input sequence $\mathbf{x}$, the pre-trained language model $f_{\boldsymbol{\theta}}$ can iteratively generate an output sequence $\hat{\mathbf{x}}$ during inference by sampling $\hat{x}_{t+1} \sim \mathbf{p}_{\boldsymbol{\theta}}(.|\hat{\mathbf{x}}_{<t})$ with $\hat{x}_t$ a generate token.

## 2.2  Knowledge Distillation Framework

In knowledge distillation (KD), the objective is to guide the learning of a student model on a specific task using a complex teacher model (Buciluundefined et al., 2006; Hinton et al., 2015) that possesses generic knowledge. This paradigm includes two main components: a cross-entropy loss ($\mathcal{L}CE$), which ensures the student model accurately predicts the gold tokens, and a secondary loss ($\mathcal{L}KD$) tasked to align the probability distributions of the student model with those of the teacher model. This alignment transfers broader knowledge not encapsulated in gold vectors (similar to standard basis vectors) used in the cross-entropy loss. Formally, the goal of the student is to minimize $\mathcal{L}$:

$$\mathcal{L} = \mathcal{L}_{CE} + \lambda \times \mathcal{L}_{KD} \tag{1}$$

where $\lambda \in \mathbb{R}^+$ can be used to control the trade-off between learning exclusively from text and knowledge coming from the teacher.

## 2.3  Knowledge Distillation Related Work

Building on Eq. 1, various cases have been examined.

**Distillation from teacher-generated text:**  Distillation from teacher-generated text occurs when $\lambda = 0$. This strategy is particularly advantageous when dealing with synthetic data (Kramchaninova & Defauw, 2022; Du et al., 2023; Ushio et al., 2023), a fact highlighted by the effectiveness of instructing large language models such as GPT-3.5/4 (Wu et al., 2023; Bubeck et al., 2023). Distillation from teacher-generated text (Kim & Rush, 2016; He et al., 2023; Hsieh et al., 2023; Zhou & Chiam, 2023) will be considered as a baseline throughout the paper. The primary drawback of these methods lies in their failure to fully leverage all the information that can be provided by the teacher.

**White-box approach:**  A further refinement of Eq. 1 occurs when $\mathcal{L}_{KD}$ relies on the internal features of the teacher to transfer knowledge(Jiao et al., 2020; Sun et al., 2020). Popular features include transformer attention and internal layers within both encoder-only and encoder-decoder models (Raman et al., 2023; Wang et al., 2020a; 2021). However, these methods require access to models' internal mechanisms, which are not available through API access, and assume similarities in architectural patterns between models, forcing adjustments for each situation.

**Black-box approach:**  In the black-box approach, practitioners only used the output logits of the model. They use these logits to align the student's output probabilities with those of the teacher through Kullback–Leibler divergence (KL) (Sanh et al., 2020). This method has emerged as one of the most widely adopted approaches, successfully distilling encoder, decoder, or encoder-decoder models (Timiryasov & Tastet, 2023; Mohammadshahi et al., 2022; Zhao et al., 2023a). However, employing KL divergence necessitates that both student and teacher share the same vocabulary, a requirement impractical with current large language models (LLMs) as reported in Fig. 1. We dig into the limitations of this method in the next section.

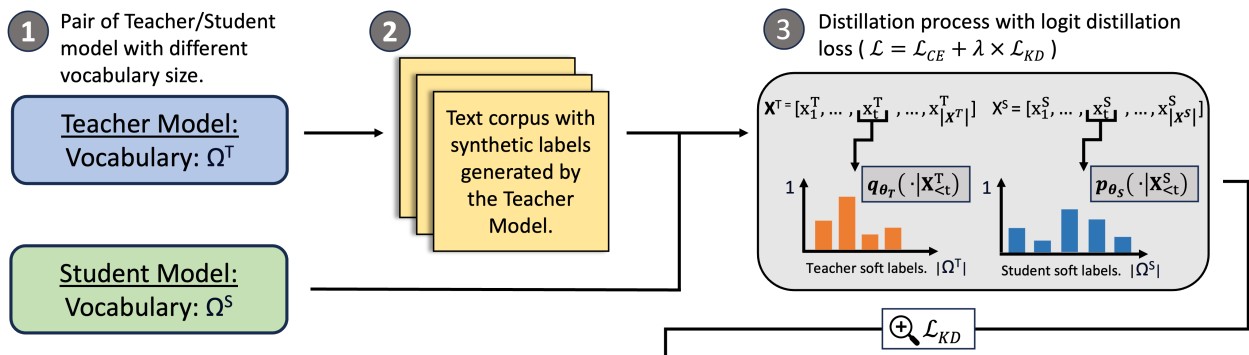

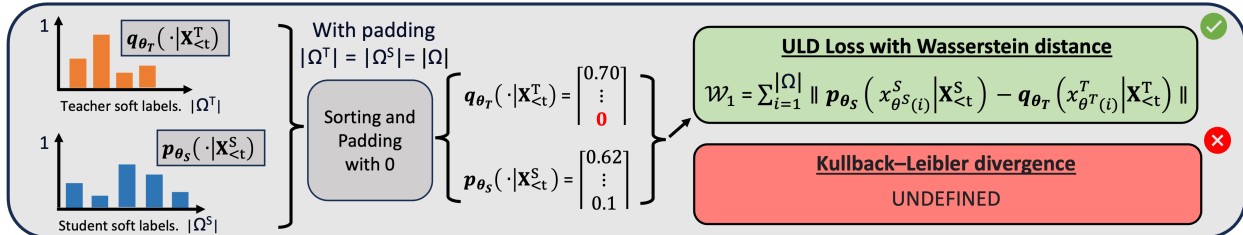

Figure 2: **Distillation using ULD loss.** In block 4, the KL divergence cannot be defined as the two distributions do not have the same support, breaking the absolute continuity of the quotient in the KL logarithmic term. To alleviate this we rely on the ULD loss which leverages a closed form of the Wasserstein distance.

## 2.4 KL Distillation loss

When distilling using the KL (Li et al., 2021), the goal is to force the student to learn the teacher's output probability distribution at each generation step. The formal definition of the objective function is provided in Eq. 2.

$$\mathcal{L} = \sum_{t=1}^{|\mathbf{x}|} \mathrm{CE}(t) + \lambda \mathrm{KL}\left[\mathbf{q}_{\boldsymbol{\theta_T}}(\cdot|\mathbf{x}_{<t}), \mathbf{p}_{\boldsymbol{\theta_S}}(\cdot|\mathbf{x}_{<t})\right] \tag{2}$$

where $|\mathbf{x}|$ the length of the tokenized sentence $\mathbf{x}$, $\mathbf{q}_{\boldsymbol{\theta_T}}(\cdot|\mathbf{x}_{<t})$ and $\mathbf{p}_{\boldsymbol{\theta_S}}(\cdot|\mathbf{x}_{<t})$ are the probability distribution for the Teacher and Student models at the $t^{th}$ steps. $\mathrm{CE}(t)$ denotes the cross-entropy loss at each generation time step $t$, expressed as: $\mathrm{CE}(t) = -\log\left(\mathbf{p}_{\boldsymbol{\theta_S}}(x_t|\mathbf{x}_{<t})\right)$ with $x_t$ the gold token for the $t^{th}$ steps and KL the Kullback–Leibler divergence defined as: $\mathrm{KL}\left[\mathbf{q}_{\boldsymbol{\theta_T}}(\cdot|\mathbf{x}_{<t}), \mathbf{p}_{\boldsymbol{\theta_S}}(\cdot|\mathbf{x}_{<t})\right] = \sum_{x\in\Omega} \mathbf{q}_{\boldsymbol{\theta_T}}(x|\mathbf{x}_{<t}) \times \log\left(\frac{\mathbf{q}_{\boldsymbol{\theta_T}}(x|\mathbf{x}_{<t})}{\mathbf{p}_{\boldsymbol{\theta_S}}(x|\mathbf{x}_{<t})}\right)$ where $\lambda \in \mathbb{R}^+$ controls the trade-off between the two terms.

**Remark.** *Eq. 2 relies on equality across the vocabulary of the student and the teacher, i.e. $\Omega = \Omega^S = \Omega^T$, ensuring similar support of probability distributions.*

**Remark.** *Eq. 2 also suppose absolute continuity for the distributions $\mathbf{p}_{\boldsymbol{\theta_S}}(\cdot|\mathbf{x}_{<t}) \ll \mathbf{q}_{\boldsymbol{\theta_T}}(\cdot|\mathbf{x}_{<t})$ at each generation step $t$, making the use of padding impractical.*

Our examination of Eq. 2 illustrating in Fig. 2 highlights that both vocabulary (similar support of probability distributions) and absolute continuity constraints pose challenges to distilling two distinct LLM families with logits using KL loss. In the following section, we introduce our ULD loss, providing a flexible framework for knowledge distillation across a wide range of LLMs.

# 3 Universal Logit Distillation

## 3.1 Background on Optimal Transport

Optimal transport provides a mathematical framework for transferring mass between distributions while minimizing cost (Villani et al., 2009; Peyré et al., 2019). Within this framework, the Wasserstein distance, also known as the Earth Mover Distance, is a robust metric for quantifying dissimilarities between distributions. This distance metric has gained popularity in NLP applications such as hallucination detection (Guerreiro et al., 2022; Shuster et al., 2021), clustering (Zhuang et al., 2022; Ye et al., 2017; Xu et al., 2018) or sentence similarity (Colombo et al., 2021; Xu et al., 2018; Bahuleyan et al., 2018).

**Wasserstein distance:** The Wasserstein distance minimizes transport costs between sampled points from all possible couplings. Let us consider two sets of probability distributions, $\mathcal{P}(\Omega_S)$ and $\mathcal{P}(\Omega_T)$, respectively containing discrete distributions over spaces $\Omega_S$ and $\Omega_T$. We denote $\mathbf{p} \in \mathcal{P}(\Omega_S)$ and $\mathbf{q} \in \mathcal{P}(\Omega_T)$ as two discrete probability distributions with $\sum_{i=1}^{|\Omega_S|} p_i \delta_{x_i^S} = 1$ and $\sum_{i=1}^{|\Omega_T|} q_i \delta_{x_i^T} = 1$, where $\delta_{x_i^S}$ and $\delta_{x_i^T}$ represent probability mass points at $x_i^S$, $x_i^T$ with $x_i^S \in \Omega_S$ and $x_i^T \in \Omega_T$ for distributions $\mathbf{p}$ and $\mathbf{q}$. The values $p_i$ and $q_i$ are weight factors ensuring that the sum of weights is equal to 1. Under this discrete setting, computing the Wasserstein distance is defined as:

$$\mathcal{W}_p(\mathbf{p}, \mathbf{q}) = \min_{T \in \Pi(\mathbf{p}, \mathbf{q})} \sum_{i=1}^{|\Omega_S|} \sum_{j=1}^{|\Omega_T|} T_{ij} C_{ij}^p \tag{3}$$

where $\Pi(\mathbf{p}, \mathbf{q})$ is the set of joint distributions with marginals $\mathbf{p}$ and $\mathbf{q}$, $C_{ij}^p$ represents the cost matrix, and $T$ is the transport plan. For the rest of the work, we focus on the Wasserstein distance related to $p = 1$. Consequently, the Wasserstein distance seeks the optimal approach to transfer probability mass from $\mathbf{p}$ to $\mathbf{q}$, minimizing the transportation cost defined by the absolute norm.

**Remark.** *Note that the Wasserstein distance (see Eq. 3) makes no assumptions about the support of $\mathbf{p}$ or $\mathbf{q}$, unlike the KL divergence, making it a natural choice for distillation.*

## 3.2 Universal Logit Distillation loss

The Universal Logit Distillation loss (ULD loss) is a novel distillation technique designed to virtually distill any generative model teacher into any student. It aims to overcome the limitations of KL divergence, as discussed in Sec. 2.4 and Fig. 2.

**Intuition.** The ULD loss retains the CE loss term to guide the model in generating the target token and introduces a Wasserstein Distance term to transfer knowledge from the teacher to the student. By minimizing the distance between the soft probabilities of the teacher and the student, our goal is to reproduce not only the predictions for the gold token but also the near-zero labels, which are crucial for performance and generalization.

**ULD loss:** Formally, the ULD loss function is formulated as:

$$\mathcal{L}_{\text{ULD}} = \sum_{t=1}^{|\mathbf{x}|} \text{CE}(t) + \lambda \times \mathcal{W}_1 \left[ \mathbf{p}_{\boldsymbol{\theta_S}} \left( \cdot | \mathbf{x}_{<t}^S \right), \mathbf{q}_{\boldsymbol{\theta_T}} \left( \cdot | \mathbf{x}_{<t}^T \right) \right] \tag{4}$$

where $|\mathbf{x}| = min\left(|\mathbf{x}^S|, |\mathbf{x}^T|\right)$ the minimum length between the sentences tokenized with the teacher or student tokenizers. Respectively $\mathbf{q}_{\boldsymbol{\theta_T}}\left(\cdot | \mathbf{x}_{<t}^T\right)$ and $\mathbf{p}_{\boldsymbol{\theta_S}}\left(\cdot | \mathbf{x}_{<t}^S\right)$ are the probability distribution for the Teacher and Student models at the $t^{th}$ steps, $\text{CE}(t)$ the cross-entropy loss defined in Sec. 2.4 and $\mathcal{W}_1$ represents the discrete Wasserstein distance defined in Eq. 3 where $\lambda \in \mathbb{R}^+$ controls the trade-off between the two terms and set to 1.5 in the rest of this paper as discussed in Appendix: Sec. B.1.

**Explanation.** Similar to the KL loss, the discrete Wasserstein distance ensures that the confidence of the student at each time step is close to the one from the teacher.

### 3.3 Fast Computation & Approximations

To the best of our knowledge, we are the first to motivate and propose the Wasserstein distance as a learning loss for distillation in the scope of the LLM decoder. Prior efforts focus on Sinkhorn regularization (Cuturi, 2013) with encoder-decoder for classification (Bhardwaj et al., 2022), while our focus diverges as we concentrate on the generative setting. This shift presents inherent challenges, as the naive computation of the Wasserstein loss in Eq. 3 exhibits a complexity of $\mathcal{O}(n^3 \log n)$, where $n$ signifies the size of the larger support. While manageable in small classification scenarios with encoders, the magnitude of the vocabulary, which can extend to 100K tokens in generative tasks, renders this approach intractable, particularly for long sequences.

**Closed form solution for ULD loss:** To achieve efficient computation of the Wasserstein distance in Eq. 4, we introduce two additional refinements:

*Uniform Support Length:* We augment either the student or teacher vocabulary size through distribution padding, ensuring equal support size for both (i.e., $|\Omega_t| = |\Omega_s| = |\Omega|$).

*Uniform Cost:* As teacher and student supports differ, and no vocabulary relationship is established, we assert that each transport cost is equal to 1. While this may seem a strong assumption, we will demonstrate that the approximation we draw still achieves better results in our case.

Under this assumption the Wasserstein distance used in the $\mathcal{L}_{\mathrm{ULD}}$ loss becomes the one introduced by Peyré et al. (2019):

$$\mathcal{W}_1 = \sum_{t=1}^{|\mathbf{x}|} \sum_{i=1}^{|\Omega|} \left| \mathbf{p}(x_{\sigma^S(i)}^S | \mathbf{x}_{<t}^S) - \mathbf{q}(x_{\sigma^T(i)}^T | \mathbf{x}_{<t}^T) \right| \tag{5}$$

with a complexity of $\mathcal{O}\left(n \cdot \log(n)\right)$ (Sec. A.1) where $\sigma^S$ and $\sigma^T$ are the permutations that sort in decreasing order the probability of student and teacher probability vectors. An extensive demonstration of this closed form is provided in App. A, written clearly and accessible.

**Intuition.** The final version of Eq. 5 is decomposed into two sums:

- The **outer sum** iterates over the sequence length (denoted as $|\mathbf{x}|$) to compute the Wasserstein distance at each time step $t$.
- The **inner sum** $\sum_i |p_{\sigma^S(s_i)} - q_{\sigma^T(t_i)}|$ represents the Wasserstein ($\mathcal{W}_1$) distance between the logits of the student and teacher models at that time step.

To compute the $\mathcal{W}_1$ distance efficiently, we use a closed-form solution derived under certain assumptions: uniform support length and uniform transport cost, as detailed in App. A. This solution computes the absolute difference between the sorted probability masses of the teacher and student vectors at time step $t$, ensuring that the overall structure of the teacher's probability distribution is preserved.

This process allows the student model to learn not only from the golden tokens (as used in the Cross-Entropy Loss) but also from the surrounding probability distribution, capturing distributional information from the teacher's predictions.

## 4 Experimental Setting

### 4.1 Evaluation Scenarios

We avoid fine-tuning teacher models to ensure alignment with the black-box approach as training teacher models may be unavailable. In this way, we enable ULD Loss to operate in an unsupervised environment by generating all *answers* text with teacher models. For repeatability and fair comparison between experiments, we opted to retain original answers for the test set split. We investigated various scenarios to evaluate the ULD loss performance across different datasets and tasks. These comprised 2 Extractive QA (Ext.), 2 Generative QA (Gen.), and 1 Summary (Sum.) tasks:

- **SQuAD (Ext.):** The Stanford Question Answering Dataset (SQuAD) (Rajpurkar et al., 2016) is a reading comprehension dataset with 87,600 questions generated by crowdworkers from Wikipedia articles. Answers are text portions from the relevant sections of the articles.

- **QED (Ext.):** The QED (Lamm et al., 2020) dataset, expertly annotated, extends from a subset of the Google Natural Questions dataset, comprising 7,640 question-answering pairs with explanations. Our focus is exclusively on extracted answers (spans).

- **FairytaleQA (Gen.):** The FairytaleQA Dataset (Xu et al., 2022), created by educational experts, consists of 10,580 questions from 278 children-friendly stories. Questions may be explicit or implicit.

- **PubMedQA (Gen.):** The PubMedQA (Jin et al., 2019) dataset contains question-answer pairs extracted from medical papers. Questions are based on titles, context on abstracts, and responses on conclusions. Due to the dataset size and context length of our student models, we subset the dataset by selecting the first 50,000 smaller items.

- **DIALOGSum (Sum.):** DialogSum (Chen et al., 2021) is a large-scale dialogue summarization dataset, consisting of 13,460 spoken dialogues with corresponding summaries and topics.

## 4.2 Experimental Choices

**Baseline:** As far as we know, the only method currently capable of distilling any pair of teacher and student LLM models in a black-box approach is distillation from teacher-generated text seen in Sec. 2.3. Throughout the remainder of this paper, distillation from teacher-generated texts will serve as the baseline for evaluating the distillation process using the ULD loss across different teacher-student pairs, tasks, and datasets. Additional experimentation with KL Div methods between models from similar families can be found in Appendix Sec. B.2

**Teacher Models:** We employed two teacher decoder models, each with 7 billion parameters: LLama 2 7b Chat (LLama) by Touvron et al. (2023b) and Mistral 7b Instruct (Mistral) by Jiang et al. (2023). These instruct models were chosen for their ability to generate few-shot answers (Brown et al., 2020; Wang et al., 2020b) across diverse tasks and their distinct vocabulary set as shown in Figure 1.

**Student Models:** We chose student models from various LLM families and architectures with parameters ranging between 160 million to 1 billion: OPT 350m (Zhang et al., 2022), Pythia 160m, Pythia 410m, Pythia 1b (Biderman et al., 2023), Bloomz 560m (Muennighoff et al., 2023) all decoder models and MT0 580m (Muennighoff et al., 2023) an encoder-decoder. It's important to note that models can have been already pre-trained on some datasets such as SQuAD for Bloomz and MT0.

**Training process:** ULD loss distillation and teacher-generated text distillation were processed uniformly. The two teacher models generate answers in inference mode for the five datasets. These answers are then utilized to train student models. During training, student models are trained exclusively to predict answers, in teacher forcing configuration. Logits used for the ULD loss are calculated by applying teacher models to the same data points they generated. Teacher's weights were frozen, ensuring consistency in teacher-generated sentences during inference and training. Additional parameter details (learning rate, batch size, etc.) can be found in the Appendix App. G.

## 4.3 Teacher Performances

Distilling using synthetic teacher-generated *answers* might restrict student performance on teacher's ones. To measure distillation efficiency accurately, we report the average native performances across tasks for both teachers Tab. 1 (details in Appendix App. D). We chose a primary metric for each task reflecting associate performances: F1 score for Extractive QA (Sokolova et al., 2006), BERTScore for Generative QA (Zhang* et al., 2020), and Rouge-Lsum for summary task (Lin, 2004). Comprehensive evaluation methods and outcomes, encompassing prompts and few-shot examples, are provided in the Appendix App. E.

Table 1: **Average performance of teacher models** across tasks with their main metrics. It is important to note a relative difference of 30% in performance between teacher models on the summary task.

| Model | Extractive (F1) | Generative (BERTScore) | Summary (Rouge-Lsum) |
|---|---|---|---|
| LLama | **69.51** | **36.11** | 23.90 |
| Mistral | 64.66 | 33.47 | **34.71** |

## 5 Empirical Results

Table 2: **Overall performance** of Teacher/Student pair models trained with ULD Loss and teacher-generated text (Raw Text) across tasks with their main metrics. Evaluations are performed over respective test splits.

| Teacher | Model | Method | SQUAD (F1) | QED (F1) | FairytaleQA (BERTScore) | PubMedQA (BERTScore) | DIALOGSum (Rouge-Lsum) |
|---|---|---|---|---|---|---|---|
| Teacher | LLama | - | **81.30** | **57.72** | **41.59** | 30.62 | 23.90 |
| | Mistral | - | 76.31 | 53.01 | 36.01 | **30.93** | **34.71** |
| LLama | OPT-350m | Raw Text | 70.78 | 48.64 | **33.78** | 27.99 | **20.58** |
| | | ULD Loss | **72.97** | **49.06** | 33.03 | **30.01** | 20.11 |
| | Pythia-410m | Raw Text | 71.39 | 47.04 | 33.02 | 29.86 | 20.94 |
| | | ULD Loss | **74.14** | **49.15** | **34.83** | **29.89** | **22.19** |
| | Bloomz-560m | Raw Text | 73.54 | 50.99 | 36.70 | 29.14 | 20.01 |
| | | ULD Loss | **75.90** | **55.33** | **37.86** | **30.01** | **22.67** |
| Mistral | OPT-350m | Raw Text | 71.64 | 50.13 | 30.09 | 27.91 | 31.44 |
| | | ULD Loss | **73.35** | **50.88** | **30.44** | **30.30** | **32.17** |
| | Pythia-410m | Raw Text | 71.50 | 47.07 | 31.44 | 28.25 | 31.64 |
| | | ULD Loss | **73.64** | **50.38** | **31.79** | **29.55** | **33.10** |
| | Bloomz-560m | Raw Text | 73.34 | 52.15 | 32.64 | 28.87 | 31.95 |
| | | ULD Loss | **76.00** | **55.79** | **33.93** | **30.60** | **32.58** |
| Average | - | Raw Text | 72.03 | 49.34 | 32.94 | 28.67 | 26.09 |
| | - | **ULD Loss** | **74.33** | **51.77** | **33.65** | **30.06** | **27.14** |

### 5.1 General Results

We empirically validate the effectiveness of the ULD loss step-by-step. First, we report in Tab. 2 the aggregated key metrics performance over the different datasets and teacher/student pairs. **ULD loss achieves the best overall results**, which indicates that the proposed ULD loss effectively improves the performances of every student model on a variety of downstream tasks using any Teacher. Notably, ULD loss exhibits an average improvement of 2.30 points over models trained on teacher-generated text for extractive QA tasks and Bloomz outperforms his teacher Mistral on the QED datasets. Furthermore, concerning summarization tasks, the 30% performance disparity between LLama/Mistral (Tab. 1) persists in their distilled counterparts (Tab. 2), underscoring the critical role of teacher performances.

### 5.2 Student Size Ablation Study

General results in Tab. 2 show a consistent pattern regarding the model size and the gain achieved with the ULD loss, especially for challenging tasks such as generative QA. To understand the impact of student size on distillation capability, we performed an ablation study over the Pythia family. We hold the training dataset size fixed at 100% and compare the performance of models from 160m, 410m to 1b parameters and

report results in Fig. 3. We observe that incorporating ULD loss consistently enhances student models across

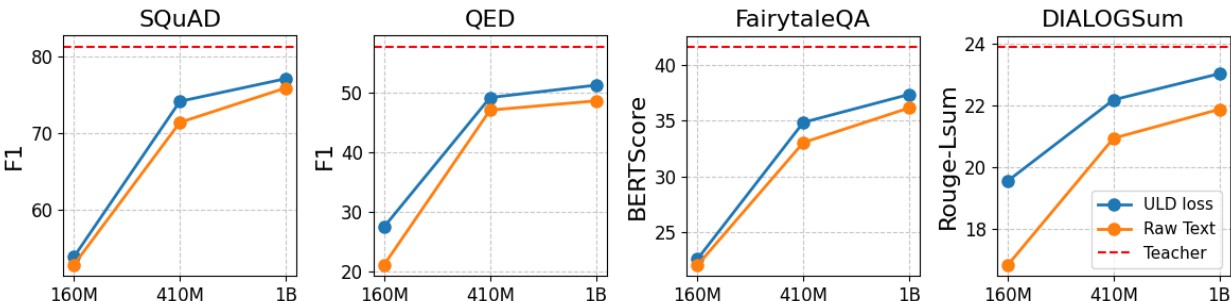

Figure 3: **Student model size ablation** with the Pythia family trained by a LLama teacher. Trainings are conducted with ULD loss and teacher-generated text (raw text). Evaluation scores on test sets are depicted on the Y-axis, while Pythia model sizes are on the X-axis.

various tasks. The enhancements are particularly noticeable for smaller models on simpler tasks, while ULD loss requires larger models for effectively distilling teacher logits on harder tasks. This is especially evident in tasks requiring reasoning, such as FairytaleQA. While using logits teacher improves training, deep reasoning tasks still require appropriate model sizes to process complex relationships taught by teachers. Generally, we observe a significant increase in capacity transfer from teacher to student models through **the use of ULD, enabling student models to match models twice bigger trained with the teacher-generated text method.** For example, Pythia 410m with ULD loss matches the performance of the Pythia 1b distilled with teacher-generated text on QED and DIALOGSum.

### 5.3 Dataset Size Ablation Study

In this section, we investigate and report in Fig. 4 the influence of the dataset size for models trained with ULD Loss or teacher-generated text. We perform ablations with respectively 25%, 50%, 75%, and 100% of dataset size while keeping training parameters constants. Training for each ratio was conducted five times to establish a range of performance. For every ablation ratio, models trained with ULD loss achieved better performance than models trained on teacher-generated text. Specifically, **with 50% of a dataset, ULD loss models overpass the performance of teacher-generated text models trained with full dataset**.

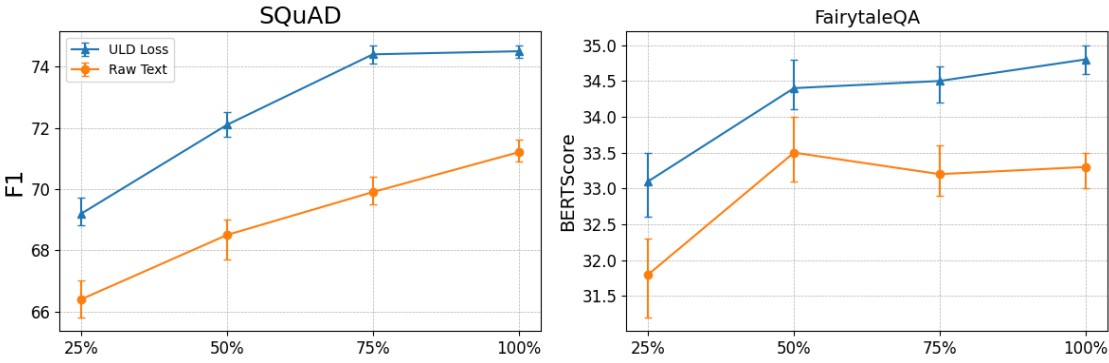

Figure 4: **Dataset size ablation** with a LLama/Pythia-410m pair trained with ULD loss or teacher-generated text. The X-axis indicates the % of data used during training while the y-axis represents the test set score. Minimum and maximum values are represented by the error bars in the plot while the mean is represented by points.

### 5.4 Training Regularization

To understand the impact of the ULD loss during training we decide to compute the validation ULD and Cross-entropy loss values for two pairs of teacher/student on the SQuAD dataset every 200 steps during 5 epochs. We report the curves formed by this point in Fig. 5. It appears that using the **ULD loss contributes to stabilizing the distillation process over training** and mitigates overfitting issues, enabling the model to train more effectively across multiple epochs. It's worth noting that incorporating the ULD loss during training stabilizes both ULD and Cross-entropy loss.

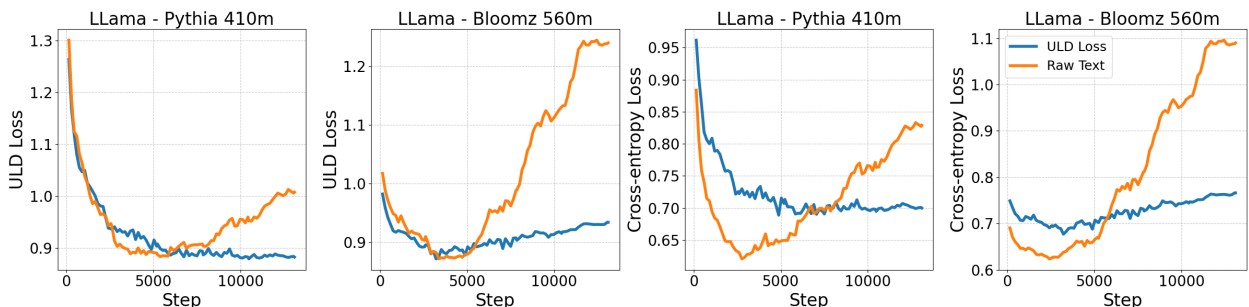

Figure 5: **Evolution of validation ULD and Cross-entropy loss curves during training** on SQuAD dataset for a LLama/Pythia-410m and LLama/Bloomz-560m Teacher/Student pair of model. For teacher-generated text models (raw text), the ULD loss was only computed during validation and did not impact the training.

**Remark.** *Even though the cross entropy is lower with raw text data than with ULD Loss, the raw data does not lead to better results. In fact, during training, models distilled with ULD Loss are trained to reproduce the predictions of the teacher model. Consequently, they are not trained to minimize cross-entropy and have maximum confidence in each prediction.*

*For this reason, a student model trained to reproduce the confidence of a teacher model of 0.92, compared to a model trained only on raw text and trained to predict 1, will have a lower probability on the golden token and consequently a higher Cross Entropy Loss. However, this model will be better calibrated to its true confidence for the sentence.*

## 6 Distillation of Decoder Teacher to Encoder-Decoder Student

As shown in Sec. 5, ULD loss effectively transfers knowledge from any pair of teacher/student decoders. Moreover, by leveraging solely on logit information and adopting a black-box approach, ULD loss should be able to extend its versatility and improve cross-architecture distillation. To validate this, we distill a teacher/student pair LLama/MT0-580m and focus our experimentation on PubMedQA, DIALOGSum, and QED to avoid any data seen during the pre-training of MT0 with the xP3 dataset (Muennighoff et al., 2023; Sanh et al., 2022).

Table 3: **Distillation of a LLama teacher (decoder) to an MT0-580m (encoder-decoder)** with ULD Loss and teacher-generated text on three data sets.

| ULD loss | QED (F1) | PubMedQA (BERTScore) | DIALOGSum (Rouge-Lsum) |
|---|---|---|---|
| Raw Labels | 55.63 | 27.56 | 23.22 |
| ULD Loss | **56.01** | **30.19** | **23.92** |

The results presented in Tab. 3 demonstrate that incorporating logit information from a decoder teacher using ULD loss can enhance the performance of an encoder-decoder student model. Notably, the inherent ability of the encoder-decoder in the summary task seems to be limited by the synthetic *answers* as teacher-generated text distillation matches the teacher's performance. However, by using the logit information with the ULD loss, the student model still leads to improved results, suggesting a successful knowledge transfer through logits. With this additional knowledge, the student model slightly outperforms the teacher one. Furthermore, in generative tasks where decoder architectures perform, the encoder-decoder student model gained 2.63 points over distillation with teacher-generated text.

## 7    Conclusions

In this work, we introduce the Universal Logit Distillation (ULD) loss, a novel approach for distilling any decoder teacher model into any student model for LLM generative tasks, utilizing a new closed form of the Wasserstein distance never seen before for distillation. ULD achieves superior overall results and matches the performance of traditional teacher-generated text distillation with only half the training dataset or student model size, while effectively preventing overfitting. Our comprehensive experiments demonstrate the efficacy of the ULD loss across a variety of tasks, datasets, and model architectures, showcasing its advantages over standard teacher-generated text distillation methods.

### Broader Impact Statement

Knowledge distillation aims to reduce the size, cost, and energy consumed by a model at inference time. Our work opens new perspectives in this area, aligned with the desire for sobriety, notably for environmental reasons. Although KD allows partial transfer of larger model performance, smaller models remain limited in their reasoning capacity and are more susceptible to hallucinatory behavior (Rawte et al., 2023), especially in open-ended generation tasks. This phenomenon has not been extensively studied in this work. Furthermore, by distilling knowledge from existing models, if a bias is already present in the teacher model, it may be transferred to the student model. This is not unique to our method, but it's a common risk for all knowledge distillation.

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

# A    Appendix - Proof of the closed Form

In this section, we gather an extensive proof of the close form used. Although the proof is not hard, we could not find it properly written in an easy and readable manner.

### Step 1: Wasserstein Distance Definition

Given two discrete distributions $\mathbf{p}$ and $\mathbf{q}$ over a vocabulary $\Omega$ of size $n$, the Wasserstein distance $\mathcal{W}_1$ is given by:

$$\mathcal{W}_1(\mathbf{p}, \mathbf{q}) = \min_\pi \sum_{i=1}^{n} \sum_{j=1}^{n} \pi(i,j) c(i,j)$$

where $\pi(i,j)$ is the transport plan, and $c(i,j)$ is the cost of transporting mass from $i$ to $j$.

### Step 2: Uniform Cost Assumption

Assume that the cost $c(i,j) = 1$ for all $i, j$. This reduces the Wasserstein distance to:

$$\mathcal{W}_1(\mathbf{p}, \mathbf{q}) = \min_\pi \sum_{i=1}^{n} \sum_{j=1}^{n} \pi(i,j) \cdot 1 = \min_\pi \sum_{i=1}^{n} \sum_{j=1}^{n} \pi(i,j)$$

### Step 3: Transport Plan Constraints

The transport plan $\pi(i,j)$ must satisfy:

$$\sum_{j=1}^{n} \pi(i,j) = \mathbf{p}(i) \quad \text{for all } i \qquad\qquad \sum_{i=1}^{n} \pi(i,j) = \mathbf{q}(j) \quad \text{for all } j$$

### Step 4: Closed-Form Solution Using Permutations

To derive a closed-form solution, we can utilize the assumptions and inherent structure of the problem:

1. **Uniform Cost Implication**: Given the uniform cost assumption, the cost of transporting mass from $i$ to $j$ is constant. Consequently, **the optimal transport plan minimizes costs by directly matching the masses** without any additional expenditures.

2. **Sorting and Matching**: Assume the distributions $\mathbf{p}$ and $\mathbf{q}$ are sorted in decreasing order. If they are not already sorted, we can introduce permutations $\sigma_S$ and $\sigma_T$ such that $\mathbf{p}(\sigma_S(i))$ and $\mathbf{q}(\sigma_T(i))$ are in sorted order.

3. **Simplified Transport Plan**: With the distributions sorted, the optimal transport plan $\pi(i,j)$ becomes straightforward. We can directly match $\mathbf{p}(\sigma_S(i))$ with $\mathbf{q}(\sigma_T(i))$ for corresponding indices $i$.

With this optimal transport plan, the Wasserstein distance simplifies to:

$$\mathcal{W}_1(\mathbf{p}, \mathbf{q}) = \sum_{i=1}^{n} |\mathbf{p}(\sigma_S(i)) - \mathbf{q}(\sigma_T(i))|$$

**Let's prove that the optimal transport plan is the one that directly matching the masses.**

We assume that the cost $c(i,j) = 1$ for all $i$ and $j$. This means that the cost of transporting any unit of mass from any $i$ to any $j$ is identical. To prove that the optimal transport plan under the uniform cost is one that directly matches the masses, we need to show that any deviation from direct matching does not reduce the cost.

**Step 1: Feasibility of the Direct Matching Transport Plan**

Consider the direct matching transport plan $\pi(i,j) = \mathbf{p}(i) \cdot \delta_{ij}$, where $\delta_{ij}$ is the Kronecker delta function. This means that $\pi(i,j)$ is non-zero only when $i = j$, and it directly assigns $\mathbf{p}(i)$ to $\mathbf{q}(i)$.

$$\pi(i,j) = \begin{cases} \mathbf{p}(i) & \text{if } i = j \\ 0 & \text{otherwise} \end{cases}$$

This plan satisfies the constraints because:

$$\sum_{j=1}^{n} \pi(i,j) = \sum_{j=1}^{n} \mathbf{p}(i) \cdot \delta_{ij} = \mathbf{p}(i) \quad \text{for all } i \qquad\qquad \sum_{i=1}^{n} \pi(i,j) = \sum_{i=1}^{n} \mathbf{p}(i) \cdot \delta_{ij} = \mathbf{q}(j) \quad \text{for all } j$$

**Step 2: Cost comparison**

According to the direct matching plan, the cost is:

$$\mathcal{W}_1 = \sum_{i=1}^{n} \pi(i,i) = \sum_{i=1}^{n} \mathbf{p}(i)$$

Similarly, for any alternative transport plan that satisfies the constraints, the total cost remains unchanged because the total amount of mass transported is conserved and each unit of mass incurs a uniform cost of 1:

$$\sum_{i=1}^{n} \sum_{j=1}^{n} \pi(i,j) = \sum_{i=1}^{n} \mathbf{p}(i)$$

**Conclusion**

Given that the cost under any valid transport plan is equivalent to the total mass transported, which is 1 per unit of mass due to uniform cost, the direct matching plan is optimal. Consequently, under the assumption of uniform cost, the optimal transport plan directly matches the masses without incurring additional costs. Thus, the Wasserstein distance can be approximated by summing the absolute differences between the aligned distributions, yielding the exact closed-form proposed under these assumptions:

$$\mathcal{W}_1(\mathbf{p}, \mathbf{q}) = \sum_{i=1}^{n} |\mathbf{p}(i) - \mathbf{q}(i)|$$

**A.1 Complexity**

$$\mathcal{W}_1 = \sum_{t=1}^{|\mathbf{x}|} \sum_{i=1}^{|\Omega|} \left| \mathbf{p}(x_{\sigma^S(i)}^S | \mathbf{x}_{<t}^S) - \mathbf{q}(x_{\sigma^T(i)}^T | \mathbf{x}_{<t}^T) \right| \qquad\qquad \text{CE}(t) = -\sum_{t=1}^{|\mathbf{x}|} \sum_{i=1}^{|\Omega|} \log\left( \mathbf{p}_{\boldsymbol{\theta_S}}(x_i | \mathbf{x}_{<t}) \right)$$

Closed form Wasserstein Distance (Eq. 5)

Cross Entropy (Sec. 2.4)

$$\mathrm{KL} = \sum_{t=1}^{|\mathbf{x}|} \sum_{i=1}^{|\Omega|} \mathbf{q}_{\boldsymbol{\theta_T}}(x_i|\mathbf{x}_{<t}) \times \log \left( \frac{\mathbf{q}_{\boldsymbol{\theta_T}}(x_i|\mathbf{x}_{<t})}{\mathbf{p}_{\boldsymbol{\theta_S}}(x_i|\mathbf{x}_{<t})} \right)$$

Kullback–Leibler Divergence (Sec. 2.4)

As discussed in (Sec. 3.3), we emphasize the significance of a newly closed-form Wasserstein distance for knowledge distillation, which from the base of our knowledge has not been previously seen. Indeed, the conventional approach poses challenges due to the Wasserstein high computational complexity of $\mathcal{O}(n^3 \cdot \log n)$, where $n$ signifies the size of the larger support, hindering scalability and practical adoption. However, our discovery of a closed-form solution with the ULD Loss markedly reduces this complexity to $\mathcal{O}(n \cdot \log(n))$. Precisely, in the case of a sentence of size $|X|$ and output vector of support size $|\Omega|$ (vocabulary length) the complexity cost $\mathcal{O}(|X| \cdot (|\Omega| \cdot \log(\Omega)))$ with $\log(\Omega)$ the complexity induced by sorting algorithms in general GPU frameworks [4].

---

[4]https://developer.nvidia.com/gpugems/gpugems2/part-vi-simulation-and-numerical-algorithms/chapter-46-improved-gpu-sorting

# B   Appendix - Complementary experiments

## B.1   Distillation Factor Ablation

| Model | Dataset | $\lambda = 0$ | $\lambda = 0.5$ | $\lambda = 1$ | $\lambda = 1.5$ | $\lambda = 2$ | $\lambda = 3$ |
|---|---|---|---|---|---|---|---|
| pythia-410m | FairytaleQA | 33.02 | 34.83 | **35.52** | 34.83 | 34.93 | 33.99 |
| pythia-410m | SQuAD | 71.39 | 74.33 | 74.53 | 74.14 | 74.75 | **74.91** |
| pythia-410m | Dialogsum | 20.94 | 22.68 | **23.26** | 22.19 | 21.74 | 21.24 |
| bloomz-560m | FairytaleQA | 36.70 | 36.98 | 36.95 | 37.86 | **37.96** | 37.21 |
| bloomz-560m | SQuAD | 73.54 | 75.96 | 75.95 | 75.90 | **77.23** | 76.92 |
| bloomz-560m | Dialogsum | 20.01 | 22.26 | 22.34 | **22.67** | 21.90 | 20.02 |
| opt-350m | FairytaleQA | 33.78 | **34.60** | 33.95 | 33.03 | 33.45 | 32.72 |
| opt-350m | SQuAD | 70.78 | 73.43 | 73.78 | 72.97 | **74.03** | 73.26 |
| opt-350m | Dialogsum | **20.58** | 20.30 | 20.02 | 20.11 | 19.63 | 18.94 |

Table 4: Ablation Study on the Lambda Distillation Factor with the LLama teacher. The values compared for each dataset are as follows: FairytaleQA (BERTScore), SQuAD (F1), Dialogsum (Rouge-Lsum)

Due to the prohibitive costs associated with training models across various datasets using multiple lambda values, we restricted our analysis to specific values. These values were identified through an ablation study involving three students across three distinct tasks, assessing six lambda values. A lambda value higher than 1 implies greater emphasis on the ULD Loss than on the cross-entropy term, as detailed in Eq. 1. Conversely, a lambda value of zero indicates that distillation proceeds without incorporating logit information. Results presented in Tab. 4 demonstrate that the ULD loss consistently enhances performance compared to baseline ($\lambda = 0$) in nearly all experimental scenarios. Based on these findings, we have chosen to focus on $\lambda = 0$ for baseline purposes and $\lambda = 1.5$ for the ULD Loss to provide the most general results in the remainder of this paper. **Importantly, the ablation suggests that fine-tuning $\lambda$ could enhance performance by as much as 1.5 percentage points over the general results presented in this paper**.

## B.2   Comparison between KL Divergence and ULD Loss

| Model | Dataset | Method | $\lambda = 0$ | $\lambda = 0.5$ | $\lambda = 1$ | $\lambda = 1.5$ | $\lambda = 2$ | $\lambda = 3$ |
|---|---|---|---|---|---|---|---|---|
| TinyLlama-1.1B | FairytaleQA | KL Div | 38.45 | 39.21 | 39.20 | 39.67 | 39.61 | 39.88 |
| TinyLlama-1.1B | FairytaleQA | ULD Loss | 38.45 | 38.53 | 39.37 | 39.39 | 39.75 | 39.36 |
| TinyLlama-1.1B | SQuAD | KL Div | 81.06 | 81.87 | 81.95 | 82.12 | 82.06 | 82.02 |
| TinyLlama-1.1B | SQuAD | ULD Loss | 81.06 | 81.52 | 81.90 | 82.04 | 81.72 | 81.47 |
| TinyLlama-1.1B | Dialogsum | KL Div | 24.77 | 24.36 | 24.17 | 24.17 | 24.03 | 24.07 |
| TinyLlama-1.1B | Dialogsum | ULD Loss | 24.77 | 24.97 | 24.74 | 24.08 | 24.48 | 24.85 |

Table 5: Comparison between a LLama-7b teacher distilled with TinyLlama-1.1B-intermediate-step-1431k-3T, a model based on the same architecture, using the Kullback-Leibler divergence or the ULD Loss. The values compared for each dataset are as follows: FairytaleQA (BERTScore), SQuAD (F1), Dialogsum (Rouge-Lsum). The distillation performer with a lambda value = 0 is equivalent to training with raw data.

Although ULD Loss is designed to function across different model families, we opted to evaluate it alongside the Kullback-Leibler divergence (KL Div) method within the specific environment, where both the student and teacher models share the same architecture. In our case the student model is TinyLlama and the teacher is Llama-7b. As illustrated in Tab. 5, ULD Loss demonstrates performance comparable to existing black-box distillation methods. Notably, we observed that the most significant improvements between raw text versus

logit distillation occur at identical lambda values for ULD Loss and KL divergence. This result raises the hypothesis of parallel behavior between the two methods when applied to models of the same architecture. However, ULD Loss offers the unique advantage of applicability across diverse model pairs, making it a superior alternative to KL divergence.

# C  Appendix - General Results

## C.1  Summary

| Teacher | Model | Method | Dataset | Rouge-1 | Rouge-2 | Rouge-L | Rouge-Lsum |
|---------|-------|--------|---------|---------|---------|---------|------------|
| Llama | Bloomz-560m | Raw Text | DIALOGSum | 24.71 | 10.06 | 19.99 | 20.01 |
| Llama | Bloomz-560m | ULD Loss | DIALOGSum | 28.08 | 11.68 | 22.64 | 22.67 |
| Mistral | Bloomz-560m | Raw Text | DIALOGSum | 39.85 | 15.36 | 31.92 | 31.95 |
| Mistral | Bloomz-560m | ULD Loss | DIALOGSum | 40.57 | 15.94 | 32.6 | 32.58 |
| Llama | OPT-350m | Raw Text | DIALOGSum | 25.4 | 10.48 | 20.57 | 20.58 |
| Llama | OPT-350m | ULD Loss | DIALOGSum | 23.69 | 9.76 | 20.13 | 20.11 |
| Mistral | OPT-350m | Raw Text | DIALOGSum | 39.33 | 14.97 | 31.49 | 31.44 |
| Mistral | OPT-350m | ULD Loss | DIALOGSum | 39.8 | 15.76 | 32.19 | 32.17 |
| Llama | Pythia-410m | Raw Text | DIALOGSum | 26.28 | 10.52 | 20.92 | 20.94 |
| Llama | Pythia-410m | ULD Loss | DIALOGSum | 27.29 | 11.2 | 22.17 | 22.19 |
| Mistral | Pythia-410m | Raw Text | DIALOGSum | 39.69 | 15.0 | 31.62 | 31.64 |
| Mistral | Pythia-410m | ULD Loss | DIALOGSum | 41.39 | 15.93 | 33.08 | 33.1 |
| Llama | Pythia-160m | Raw Text | DIALOGSum | 20.34 | 7.46 | 16.81 | 16.81 |
| Llama | Pythia-160m | ULD Loss | DIALOGSum | 22.94 | 8.39 | 19.56 | 19.55 |
| Llama | Pythia-1b | Raw Text | DIALOGSum | 27.71 | 11.08 | 21.86 | 21.88 |
| Llama | Pythia-1b | ULD Loss | DIALOGSum | 28.48 | 12.16 | 23.04 | 23.04 |

Table 6: **Details performance of Teacher/Student pair models trained with ULD Loss and teacher-generated text (Raw Text) for the Summary task.** Evaluations are performed over respective test splits.

## C.2  Extractive QA

| Teacher | Model | Method | Dataset | F1 | Precision | Recall |
|---------|-------|--------|---------|-----|-----------|--------|
| Llama | Bloomz-560m | Raw Text | SQuAD | 73.54 | 75.35 | 75.19 |
| Llama | Bloomz-560m | ULD Loss | SQuAD | 75.9 | 77.37 | 77.88 |
| Llama | Bloomz-560m | Raw Text | QED | 50.99 | 58.9 | 52.38 |
| Llama | Bloomz-560m | ULD Loss | QED | 55.33 | 63.22 | 56.47 |
| Mistral | Bloomz-560m | Raw Text | SQuAD | 73.34 | 73.31 | 78.52 |
| Mistral | Bloomz-560m | ULD Loss | SQuAD | 76.0 | 76.1 | 81.1 |
| Mistral | Bloomz-560m | Raw Text | QED | 52.15 | 57.49 | 56.28 |
| Mistral | Bloomz-560m | ULD Loss | QED | 55.79 | 61.98 | 58.8 |
| Llama | OPT-350m | Raw Text | SQuAD | 70.78 | 72.52 | 72.78 |
| Llama | OPT-350m | ULD Loss | SQuAD | 72.97 | 74.61 | 74.99 |
| Llama | OPT-350m | Raw Text | QED | 48.64 | 54.74 | 51.84 |
| Llama | OPT-350m | ULD Loss | QED | 49.06 | 55.38 | 51.74 |
| Mistral | OPT-350m | Raw Text | SQuAD | 71.64 | 71.67 | 77.28 |
| Mistral | OPT-350m | ULD Loss | SQuAD | 73.35 | 73.25 | 78.91 |
| Mistral | OPT-350m | Raw Text | QED | 50.13 | 55.36 | 54.56 |
| Mistral | OPT-350m | ULD Loss | QED | 50.88 | 56.61 | 54.53 |
| Llama | Pythia-410m | Raw Text | SQuAD | 71.39 | 73.76 | 72.85 |
| Llama | Pythia-410m | ULD Loss | SQuAD | 74.14 | 75.88 | 76.31 |
| Llama | Pythia-410m | Raw Text | QED | 47.04 | 54.31 | 48.87 |
| Llama | Pythia-410m | ULD Loss | QED | 49.15 | 54.75 | 53.13 |
| Mistral | Pythia-410m | Raw Text | SQuAD | 71.5 | 71.33 | 77.54 |
| Mistral | Pythia-410m | ULD Loss | SQuAD | 73.64 | 73.34 | 79.71 |
| Mistral | Pythia-410m | Raw Text | QED | 47.07 | 50.2 | 54.67 |
| Mistral | Pythia-410m | ULD Loss | QED | 50.38 | 54.19 | 56.62 |
| Llama | Pythia-160m | Raw Text | SQuAD | 52.83 | 53.68 | 56.67 |
| Llama | Pythia-160m | ULD Loss | SQuAD | 53.86 | 54.57 | 58.19 |
| Llama | Pythia-160m | Raw Text | QED | 21.11 | 21.69 | 34.46 |
| Llama | Pythia-160m | ULD Loss | QED | 27.48 | 30.3 | 33.8 |
| Llama | Pythia-1b | Raw Text | SQuAD | 75.89 | 77.36 | 78.28 |
| Llama | Pythia-1b | ULD Loss | SQuAD | 77.1 | 78.57 | 79.55 |
| Llama | Pythia-1b | Raw Text | QED | 48.59 | 51.05 | 60.41 |
| Llama | Pythia-1b | ULD Loss | QED | 51.22 | 55.3 | 59.7 |

Table 7: **Details performance of Teacher/Student pair models trained with ULD Loss and teacher-generated text (Raw Text) for Extractive QA task.** Evaluations are performed over respective test splits.

## C.3 Generative QA

| Teacher | Model | Method | Dataset | BERTScore | PBERT | RBERT |
|---|---|---|---|---|---|---|
| Llama | Bloomz-560m | Raw Text | FairytaleQA | 36.7 | 45.42 | 28.46 |
| Llama | Bloomz-560m | ULD Loss | FairytaleQA | 37.86 | 46.93 | 29.36 |
| Llama | Bloomz-560m | Raw Text | PubMedQA | 29.14 | 29.45 | 28.86 |
| Llama | Bloomz-560m | ULD Loss | PubMedQA | 30.01 | 32.5 | 27.65 |
| Mistral | Bloomz-560m | Raw Text | FairytaleQA | 32.64 | 39.46 | 26.09 |
| Mistral | Bloomz-560m | ULD Loss | FairytaleQA | 33.93 | 42.45 | 25.8 |
| Mistral | Bloomz-560m | Raw Text | PubMedQA | 28.87 | 28.59 | 29.14 |
| Mistral | Bloomz-560m | ULD Loss | PubMedQA | 30.6 | 32.08 | 29.13 |
| Llama | OPT-350m | Raw Text | FairytaleQA | 33.78 | 41.46 | 26.57 |
| Llama | OPT-350m | ULD Loss | FairytaleQA | 33.03 | 41.16 | 25.32 |
| Llama | OPT-350m | Raw Text | PubMedQA | 27.99 | 28.46 | 27.56 |
| Llama | OPT-350m | ULD Loss | PubMedQA | 30.01 | 36.11 | 24.14 |
| Mistral | OPT-350m | Raw Text | FairytaleQA | 30.09 | 35.47 | 24.91 |
| Mistral | OPT-350m | ULD Loss | FairytaleQA | 30.44 | 37.38 | 23.81 |
| Mistral | OPT-350m | Raw Text | PubMedQA | 27.91 | 27.06 | 28.75 |
| Mistral | OPT-350m | ULD Loss | PubMedQA | 30.3 | 36.99 | 23.82 |
| Llama | Pythia-410m | Raw Text | FairytaleQA | 33.02 | 41.31 | 25.26 |
| Llama | Pythia-410m | ULD Loss | FairytaleQA | 34.83 | 42.61 | 27.49 |
| Llama | Pythia-410m | Raw Text | PubMedQA | 29.86 | 31.06 | 28.72 |
| Llama | Pythia-410m | ULD Loss | PubMedQA | 29.89 | 31.23 | 28.62 |
| Mistral | Pythia-410m | Raw Text | FairytaleQA | 31.44 | 37.97 | 25.18 |
| Mistral | Pythia-410m | ULD Loss | FairytaleQA | 31.79 | 38.17 | 25.71 |
| Mistral | Pythia-410m | Raw Text | PubMedQA | 28.25 | 25.91 | 30.56 |
| Mistral | Pythia-410m | ULD Loss | PubMedQA | 29.55 | 30.09 | 29.01 |
| Llama | Pythia-160m | Raw Text | FairytaleQA | 22.03 | 30.05 | 14.5 |
| Llama | Pythia-160m | ULD Loss | FairytaleQA | 22.58 | 31.61 | 14.08 |
| Llama | Pythia-160m | Raw Text | PubMedQA | 26.54 | 26.26 | 26.85 |
| Llama | Pythia-160m | ULD Loss | PubMedQA | 29.78 | 36.4 | 23.36 |
| Llama | Pythia-1b | Raw Text | FairytaleQA | 36.13 | 46.11 | 26.74 |
| Llama | Pythia-1b | ULD Loss | FairytaleQA | 37.34 | 46.93 | 28.33 |
| Llama | Pythia-1b | Raw Text | PubMedQA | 30.12 | 31.67 | 28.64 |
| Llama | Pythia-1b | ULD Loss | PubMedQA | 29.88 | 30.4 | 29.44 |

Table 8: **Details performance of Teacher/Student pair models trained with ULD Loss and teacher-generated text (Raw Text) for generative tasks.** Evaluations are performed over respective test splits.

# D   Appendix - Native Performances

Base models used as teacher and student can be respectively download on HuggingFace: LLama 2 7b Chat[5], Mistral 7b Instruct[6], Pythia 160m, Pythia 410m, Pythia 1b[7], Bloomz 560m, MT0 580m[8].

## D.1   Summary

| Model | Dataset | Number Few-Shot | Few-Shot Titled | Rouge-1 | Rouge-2 | Rouge-L | Rouge-Lsum |
|---|---|---|---|---|---|---|---|
| Bloomz-560m | DIALOGSum | 3 | False | 15.36 | 1.47 | 11.92 | 11.9 |
| OPT-350m | DIALOGSum | 2 | False | 22.06 | 3.31 | 17.86 | 17.84 |
| Pythia-410m | DIALOGSum | 3 | False | 23.38 | 6.4 | 20.12 | 20.13 |
| Pythia-160m | DIALOGSum | 3 | False | 16.0 | 4.72 | 13.88 | 13.88 |
| Pythia-1b | DIALOGSum | 3 | False | 33.95 | 11.85 | 29.06 | 29.1 |
| Llama | DIALOGSum | 3 | False | 0.3 | 0.13 | 0.24 | 0.24 |
| Mistral | DIALOGSum | 2 | False | 0.43 | 0.18 | 0.35 | 0.35 |

Table 9: **Native performance details of Teacher/Student pair models benchmark in few-shot setting for the Summary task.** Evaluations are performed over respective test splits.

## D.2   Extractive QA

| Model | Dataset | Number Few-Shot | Few-Shot Titled | F1 | Precision | Recall |
|---|---|---|---|---|---|---|
| Bloomz-560m | SQuAD | 3 | False | 66.05 | 68.6 | 66.09 |
| Bloomz-560m | QED | 3 | False | 41.01 | 51.55 | 38.83 |
| OPT-350m | SQuAD | 3 | False | 30.01 | 29.34 | 41.04 |
| OPT-350m | QED | 3 | False | 30.21 | 32.82 | 37.6 |
| Pythia-410m | SQuAD | 3 | False | 37.4 | 36.58 | 47.55 |
| Pythia-410m | QED | 3 | False | 33.35 | 38.05 | 37.02 |
| Pythia-160m | SQuAD | 3 | False | 15.05 | 16.39 | 18.83 |
| Pythia-160m | QED | 3 | False | 15.48 | 20.15 | 17.31 |
| Pythia-1b | SQuAD | 3 | False | 48.41 | 48.52 | 55.55 |
| Pythia-1b | QED | 3 | False | 41.72 | 47.18 | 45.76 |
| Llama | SQuAD | 1 | False | 0.81 | 0.83 | 0.84 |
| Llama | QED | 5 | False | 0.58 | 0.64 | 0.63 |
| Mistral | SQuAD | 3 | True | 0.76 | 0.74 | 0.89 |
| Mistral | QED | 5 | True | 0.53 | 0.55 | 0.68 |

Table 10: **Native performance details of Teacher/Student pair models benchmark in few-shot setting for extractive QA tasks.** Evaluations are performed over respective test splits.

## D.3   Generative QA

| Model | Dataset | Number Few-Shot | Few-Shot Titled | BERTScore | PBERT | RBERT |
|---|---|---|---|---|---|---|
| Bloomz-560m | FairytaleQA | 3 | False | 27.43 | 31.42 | 23.67 |
| Bloomz-560m | PubMedQA | 3 | False | -20.3 | -9.43 | -30.97 |
| OPT-350m | FairytaleQA | 3 | False | 3.82 | -4.33 | 13.1 |
| OPT-350m | PubMedQA | 3 | False | 19.98 | 23.29 | 16.94 |
| Pythia-410m | FairytaleQA | 3 | False | 6.76 | 1.82 | 12.43 |
| Pythia-410m | PubMedQA | 3 | False | 25.65 | 30.13 | 21.42 |
| Pythia-160m | FairytaleQA | 3 | False | -0.96 | -6.87 | 6.3 |
| Pythia-160m | PubMedQA | 3 | False | 21.35 | 27.14 | 15.93 |
| Pythia-1b | FairytaleQA | 3 | False | 20.59 | 22.4 | 19.23 |
| Pythia-1b | PubMedQA | 3 | False | 26.13 | 29.29 | 23.24 |
| Llama | FairytaleQA | 2 | False | 0.42 | 0.48 | 0.36 |
| Llama | PubMedQA | 3 | False | 0.31 | 0.3 | 0.32 |
| Mistral | FairytaleQA | 5 | True | 0.41 | 0.38 | 0.45 |
| Mistral | PubMedQA | 3 | False | 0.31 | 0.28 | 0.34 |

Table 11: **Native performance details of Teacher/Student pair models benchmark in few-shot setting for generative QA tasks.** Evaluations are performed over respective test splits.

---

[5]https://huggingface.co/meta-llama/Llama-2-7b-chat-hf
[6]https://huggingface.co/mistralai/Mistral-7B-Instruct-v0.2
[7]https://huggingface.co/EleutherAI
[8]https://huggingface.co/bigscience

# E    Appendix - Few-Shot examples and Prompt Systems

The few-shot technique was used to generate synthetic data with the teacher. The number of few-shots reported for evaluating teacher models in App. D are the same numbers used to generate the synthetic answers. It's also important to note that the few-shot method was only used to determine the native performance of the teacher and student, not the distilled versions.

## E.1    Prompt Systems

List of prompt system used with teacher templates. The default templates for chat models provided with the huggingface tokenizer have been retained:

- **Extractive QA:** You are an agent answering questions as part of a reading comprehension activity. You must read and understand the context text step by step. Answers are brief and consist exclusively of continuous words taken from the context text provided.

- **Generative QA:** You are an expert agent in reading comprehension (question answering). You must read and understand the contextual text step by step, then answer the question. The answer must be brief.

- **Summary:** You're an expert at summarizing dialogues. You have to read the dialogue between two people and summarize it in no more than one sentence. The summary should be as short as possible, not re-explaining the dialogue in detail and using the person's name when implicitly mentioned.

## E.2    Few-Shot examples

| Title | Context | Question | Answer |
|---|---|---|---|
| Christine's boyfriend | Patrick Harris (Tim DeKay), Old Christine's new boyfriend, who she meets in a video store and starts dating. | Who played patrick on new adventures of old christine? | Tim DeKay |
| June 14, 2018: Death Row Inmates | As of June 14, 2018, there were 2,718 death row inmates in the United States. | Total number of death row inmates in the us? | 2,718 |
| Modern Communism | Most modern forms of communism are grounded at least nominally in Marxism, an ideology conceived by noted sociologist Karl Marx during the mid nineteenth century. | Who came up with the idea of communism? | Karl Marx |
| Napoleon's Defeat by Seventh Coalition | A French army under the command of Napoleon Bonaparte was defeated by two of the armies of the Seventh Coalition : a British-led Allied army under the command of the Duke of Wellington, and a Prussian army under the command of Gebhard Leberecht von Blücher, Prince of Wahlstatt. | Who commanded british forces at the battle of waterloo? | The Duke of Wellington |
| Canine character | Astro is a canine character on the Hanna-Barbera cartoon, The Jetsons. | What was the dog's name on the jetsons? | Astro |

Table 12: **Few-shot examples for extractive QA** used to benchmark models and generate synthetic answers from teachers.

| Context | Summary |
|---|---|
| #Person1#: John, shall we go to Sun Store? I have decided to buy that Murrberry handbag. Anyway, I'm not carrying this one to Mary's wedding. #Person2#: But, Jane, why not rent one with Handbag Hire? Instead of 990, $pay$ 50, and you have it for a whole week. #Person1#: Sounds great, but I never knew I can rent a handbag. #Person2#: Handbag Hire is a new business. It was founded two months ago. Its collection covers many designer handbags. #Person1#: So... for the price of one Murrberry, I can use a different bag each week for twenty weeks? #Person2#: Absolutely. And if you like one of them, you can choose to buy it at a discounted rate. Of course, the price varies by age and condition. For example, a $ 1500 Murrberry bag can sell for just $750. #Person1#: Great, but how do I rent? By telephone? Or in person? #Person2#: Either. And more conveniently, it accepts online orders. #Person1#: I'll do it online now. I still have one more question. Mary's wedding is next Saturday. There are only five days left. Do I have enough time? #Person2#: Don't worry. It promises that customers receive their orders by post within two days. Three more days to go. #Person1#: Oh, I'd better order one right now. | Jane wants to buy that Murrberry handbag to carry to Mary's wedding, but John suggests renting one with Handbag Hire and tells her about the service in detail. Jane is pleased to have a try. |
| #Person1#: The summers are so great here! Not hot at all. I love the cooling breezes, the clear air, all the greenery. #Person2#: This really has been a wonderful holiday for us. Shall we take a walk around the pond or into those woods for a while? #Person1#: Let's do both! Are we in a rush or anything? #Person2#: No, not really. I had thought we'd stay in Hamburg tonight, but we can't unless we rush it. Let's stay in Bremen instead. Tomorrow we can have lunch in Hamburg, then check into a hostel in Copenhagen and have dinner there. #Person1#: Sounds fine to me. Whatever, let's enjoy this pond first. #Person2#: Sure. We can walk around to that path that leads into the woods there. Hey, look! There are some wild ducks over there in the reeds. #Person1#: I see them! Wow! How do you know they're wild? #Person2#: I used to go hunting with my uncle, that's how. #Person1#: They're neat. Now let's take that path into the woods and see what we can see... | #Person1# and #Person2# are enjoying a pond. #Person1# and #Person2# had planned to stay in Hamburg tonight, but they decide to stay in Bremen since they are not in a rush. |
| #Person1#: Well, Rebecca, is there anything else you need to know for now? #Person2#: I don't think so, Mr. Parsons. I think you have covered all the main points for me. #Person1#: Okay well listen, here is my business card with my mobile number. If any other questions spring to mind don't hesitate to contact me. Of course, you can also call Miss Childs too. #Person2#: Great. Rmm, when can I expect to hear from you? #Person1#: Well, we are finishing the shortlist interviews tomorrow, so we will certainly have a decision made by early next week. Miss Childs will call you to discuss more on Monday or Tuesday. How does that sound? #Person2#: That sounds perfect. Thank you very much for taking the time to speak to me Mr. Parsons. #Person1#: The pleasure's all mine, Rebecca. #Person2#: I hope to hear from you very soon. #Person1#: Absolutely. Thanks for coming Rebecca. Goodbye. | Mr. Parsons gives Rebecca his business card after the interview and tells Rebecca the decision will be made by early next week and Miss Childs will contact Rebecca. |

Table 13: **Few-shot examples for summary** used to benchmark models and generate synthetic summary from teachers.

| Title | Context | Question | Answer |
|---|---|---|---|
| The Wee Bannock | So, she jumped up with her lint and her lint cards, and the tailor jumped up with his great shears, and one apprentice grasped the line measure, while another took up the saucer full of pins; and they all tried to catch the wee bannock. But it dodged them round and round the fire, and at last it got safely out of the door and ran down the road, with one of the apprentices after it, who tried to snip it in two with his shears. It ran too quickly for him, however, and at last he stopped and went back to the house, while the wee bannock ran on until it came to a tiny cottage by the roadside. it trundled in at the door, and there was a weaver sitting at his loom, with his wife beside him, winding a clue of yarn. | How did the bannock escape from the tailor's wife and the three tailors? | Dodged them round and round the fire, and at last it got safely out of the door and ran down the road. |
| Princess Glass Mountain | Then he took the prince by the hand, led him deep down in the earth into his cave, and there on the wall hung a suit of armor altogether forged of the clearest silver, and so bright that it shone afar. Right beside it stood a snow-white steed, saddled and bridled, pawing the earth with his silver hoofs, and champing his bit till the foam dropped to the ground. The wild man said: 'now get quickly into your armor, ride out and try your luck! in the meantime I will tend your oxen.' The prince did not wait to be told a second time; but put on his helmet and armor in all haste, securely buckled on his spurs, hung his sword at his side, and felt as light in his silver armor as a bird in the air. Then he leaped into the saddle so that every clasp and buckle rang, laid his reins on the neck of his steed, and rode hastily toward the glass mountain. | What was the suit of armor given by the wild man forged from? | The clearest silver. |
| Money Box | He knew very well that he had enough inside him to buy up all the other toys, and this gave him a very good opinion of his own value. The rest thought of this fact also, although they did not express it, for there were so many other things to talk about. A large doll, still handsome, though rather old, for her neck had been mended, lay inside one of the drawers which was partly open. She called out to the others, 'let us have a game at being men and women, that is something worth playing at.' | Why didn't the other toys talk about how valuable the pig was? | There were so many other things to talk about. |
| A Legend of Confucius | When confucius came to the earth, the kilin, that strange beast which is the prince of all four-footed animals, and only appears when there is a great man on earth, sought the child and spat out a jade whereon was written: 'son of the watercrystal you are destined to become an uncrowned king!' and confucius grew up, studied diligently, learned wisdom and came to be a saint. He did much good on earth, and ever since his death has been reverenced as the greatest of teachers and masters. He had foreknowledge of many things and even after he had died, he gave evidence of this. | Why was confucius's death reverenced as the greatest of teachers and masters? | He did much good on earth. |
| Naughty Boy | 'Oh, let me in! Let me in! I'm cold, and I'm so wet!' Exclaimed suddenly a child that stood crying at the door and knocking for admittance, while the rain poured down, and the wind made all the windows rattle. 'Poor thing!' said the old poet, as he went to open the door. there stood a little boy, quite naked, and the water ran down from his long golden hair. He trembled with cold, and had he not come into a warm room he would most certainly have perished in the frightful tempest. | Why did the boy ask to come inside? | He was cold and wet. |

Table 14: **Few-shot examples for generative QA** used to benchmark models and generate synthetic answers from teachers.

| Context | Question | Answer |
|---|---|---|
| Injury severity score (ISS), Glasgow coma score (GCS), and revised trauma score (RTS) are the most frequently used methods to evaluate the severity of injury in blunt trauma patients. ISS is too complicated to assess easily and GCS and RTS are easy to assess but somewhat subjective. White blood cell count (WBC) is an easy, quick and objective test. This study was performed to evaluate the significance of the WBC count at presentation in the blunt trauma patients. 713 blunt trauma patients, who were admitted to the Uludag University Medical Center Emergency Department between 01.04.2000-31.12.2000, were retrospectively evaluated in terms of ISS, GCS, RTS and white blood cell count at presentation. Statistical analysis revealed that WBC was correlated positively with ISS, but negatively with GCS and RTS. | Does the leukocyte count correlate with the severity of injury | The leukocyte count at presentation can be used as an adjunct in the evaluation of the severity of injury in blunt trauma patients. |
| The aim of this study was to assess the diagnostic value of articular sounds, standardized clinical examination, and standardized articular ultrasound in the detection of internal derangements of the temporomandibular joint. Forty patients and 20 asymptomatic volunteers underwent a standardized interview, physical examination, and static and dynamic articular ultrasound. Sensitivity, specificity, and predictive values were calculated using magnetic resonance as the reference test. A total of 120 temporomandibular joints were examined. Based on our findings, the presence of articular sounds and physical signs are often insufficient to detect disk displacement. Imaging by static and dynamic high-resolution ultrasound demonstrates considerably lower sensitivity when compared with magnetic resonance. Some of the technical difficulties resulted from a limited access because of the presence of surrounding bone structures. | Internal derangement of the temporomandibular joint: is there still a place for ultrasound? | The present study does not support the recommendation of ultrasound as a conclusive diagnostic tool for internal derangements of the temporomandibular joint. |
| Figures from the British Defence Dental Services reveal that serving personnel in the British Army have a persistently lower level of dental fitness than those in the Royal Navy or the Royal Air Force. No research had been undertaken to ascertain if this reflects the oral health of recruits joining each Service. This study aimed to pilot a process for collecting dental and sociodemographic data from new recruits to each Service and examine the null hypothesis that no differences in dental health existed. Diagnostic criteria were developed, a sample size calculated and data collected at the initial training establishments of each Service. Data for 432 participants were entered into the analysis. Recruits in the Army sample had a significantly greater prevalence of dental decay and greater treatment resource need than either of the other two Services. Army recruits had a mean number of 2.59 (2.08, 3.09) decayed teeth per recruit, compared to 1.93 (1.49, 2.39 p<0.01) in Royal Navy recruits and 1.26 (0.98, 1.53 p<0.001) in Royal Air Force recruits. Among Army recruits 62.7% were from the two most deprived quintiles of the Index of Multiple Deprivation compared to 42.5% of Royal Naval recruits and 36.6% of Royal Air Force recruits. | Is there a differential in the dental health of new recruits to the British Armed Forces? | A significant difference in dental health between recruits to each Service does exist and is a likely to be a reflection of the sociodemographic background from which they are drawn. |

Table 15: **Few-shot examples for generative QA** used to benchmark models and generate synthetic answers from teachers for medical topic.

## F  Appendix - Non-uniform transport costs Systems

We study how the uniform transport cost affects ULD Loss by comparing distillation results with and without it, as shown in Tab. 16. To emphasize the computational cost of the Wasserstein distance with non-uniform transport costs, which scales as $\mathcal{O}(n^3 \log n)$ with $n$ being the size of the larger support, we use teacher and student models with both small and large vocabulary sizes. For small vocabularies, we select LLaMA ($32,000$ tokens) as the teacher and Pythia-410 ($50,304$ tokens) as the student. For large vocabularies, we pair LLaMA with Bloomz-560m ($250,880$ tokens). We use $5k$ samples of the SQuAD dataset due to its short answers, averaging 5.52 tokens for LLaMA, 3.48 for Pythia, and 3.37 for Bloomz. For non-uniform transport costs, the Levenshtein distance and the L2 norm between embedding tokens are used to define the cost of transporting mass between points in the transport plan.

**The Levenshtein distance** (Lcvenshtcin, 1966) measures how different two strings or tokens are. This distance is the smallest number of operations needed to turn one string into another. These operations are: inserting a character, deleting a character, or substituting a character, each increasing the cost of transformation by 1. For example, converting "cat" to "cats" needs one insertion, giving a distance of 1.

Before distillation training, we calculated the distance between all teacher and student tokens, resulting in a matrix $C$ of size ($|\Omega^t|, |\Omega^s|$) with $|\Omega^t|$ the teacher vocabulary size and $|\Omega^s|$) the student vocabulary size. Each element $C_{ij} = C_{ji}$ represents the transport cost between points $i$ and $j$ in the transport plan. This matrix was used as a cost dictionary during training in the Wasserstein distance computation Eq. 3.

**Embedding token** (Bojanowski et al., 2017) is a process that maps tokens (such as words or subwords) into continuous vector representations in a high-dimensional space. In this space, tokens with similar meanings or contexts are positioned closer together, reflecting their semantic similarity. We use the L2 norm between the teacher and student token embeddings to compute the cost matrix. However, since the teacher and student models do not share a common embedding space, we employ the widely used FastText subword embeddings [9] as a shared space. To achieve this, each token from the teacher and student models is mapped to its nearest token in the FastText vocabulary based on the Levenshtein distance. The corresponding FastText embedding is then assigned to the respective teacher or student token. This approach allows us to compute a matrix of L2 norm distances between all teacher and student token embedding representations. This distance matrix is then used as a cost dictionary during training in the Wasserstein distance computation.

To compute the Wasserstein distance during training, we used the POT library (Flamary et al., 2021). Extensive GPU memory management including, Offloading Optimizer/Gradient state states to CPU, activation offloading, and gradient accumulation, were employed to address the high memory demands of calculating transport plans and cost matrices. We highlight, that these experiments were conducted on A100 GPUs with 80GB of memory, using SQuAD dataset with an average label length of 4.12 tokens. Longer sentences were infeasible due to severe memory constraints.

Table 16: Distillation using ULD Loss for LLama teacher to a Pythia-410m and Bloomz-560m student with and without uniform transport cost.

| Student | Transport Cost | F1 score | Training time |
|---------|---------------|----------|---------------|
| Pythia | Uniform | 69.56 | **64 minutes** ($\times 1$) |
| Pythia | Levenshtein distance | 69.32 | 172 minutes ($\times 2.68$) |
| Pythia | Embedding L2 norm | 69.61 | 203 minutes ($\times 3.17$) |
| Bloomz | Uniform | 72.11 | **59 minutes** ($\times 1$) |
| Bloomz | Levenshtein distance | 72.08 | 221 minutes ($\times 3.74$) |
| Bloomz | Embedding L2 norm | 72.37 | 265 minutes ($\times 4.49$) |

For all experiments, in our limited setting, we do not observe a consistent improvement between models trained with ULD Loss and uniform transport costs defined by the Levenshtein distance or Embedding L2

---

[9]`https://fasttext.cc/docs/en/english-vectors.html`

Norm. These methods were however slower, requiring up to $4.49\times$ the time of uniform cost training, particularly as the model vocabulary size increased, such as when distilling Llama into Bloomz. Furthermore, training with non-uniform transport costs introduced additional challenges, including pre-training steps and intensive memory management to avoid GPU "out of memory" errors. These findings emphasize the practicality of using uniform costs for maintaining training efficiency while achieving comparable performance. We encourage future work to explore novel cost matrices that balance computational complexity with potential gains in model performance.

# G   Appendix - Training Information

During training, all distillation processes were performed over 5 epochs with a batch size of 8 for the SQuAD dataset and 4 for the others. A one-cycle learning rate scheduler was used with the following configuration for decoder models: max_lr = $1e-6$, initial_lr = max_lr/2, min_lr = initial_lr/5. For mt0 (encoder-decoder), the max learning rate parameter varied according to datasets: DIALOGSum: 1e-4, PubMedQA: 3e-4, and QED: 7e-6. Finally, distillation was performed in BFLOAT16 mode introduced by Kalamkar et al. (2019), on 4*NVIDIA A100-SXM4-80GB with the Fully Sharded Data Parallel (FSDP) technique (Zhao et al., 2023b). In total 4.923 GPU hours were used (i.e. consumption for the entire project in tonnes of $CO_2$: 0.268).

