# OpenReview forum: "Towards Cross-Tokenizer Distillation: the Universal Logit Distillation Loss for LLMs"
_TMLR — Accepted by TMLR_

### Review · Reviewer_cjLP · 2024-09-18

**Summary Of Contributions:**

Existing distillation methods either train on teacher-generated text or teacher-generated logits (only if the teacher and student have the same token space). This work addresses this shortcoming by proposing a generic loss that aligns students with teachers even when the tokenizers are different by aligning the sorted logprobs of both model's next token distribution. This works better than training on teacher-generated data for fine-tuning based distillation setups.

**Audience:**

Yes

**Claims And Evidence:**

Yes

**Requested Changes:**

- Please use \citet and \citep appropriately when citing papers reflecting whether citations should go in parentheses.
- Isn't the complexity of your distance n log n (not n) because of the sorting?
- [The following experiment suggestion is not required] To evaluate how good the assumptions are in practice, one ablation could be to do distillation between a teacher and student models with the same tokenizer. Then, you could evaluate the "oracle" best performance of standard distillation and ULD which is oblivious of the token relationships. The gap between oracle, ULD, raw text, and no training could indicate how much best case performance is recovered by ULD.

**Strengths And Weaknesses:**

Strengths

1. The paper tackles an important problem without an obvious solution: logit-based distillation has more signal but requires the same tokenizer, while text-based distillation is more general but has less signal.
2. The method is exceedingly simple, enabling easy adoption within existing frameworks with near zero extra overhead
3. The paper rigorously validates the method against reasonable baselines, establishing that the method works

Weaknesses

1. Though I can see that the idea was motivated by optimal transport, the assumptions that make the problem tractable remove the need for standard considerations in optimal transport. The final method simply makes the sorted logprobs as close to each other as possible. As the paper acknowledges, this is not necessarily principled, but is redeemed by its strong empirical performance.
2. I am slightly confused by Figure 5: If we use early stopping on say the rightmost figure, can't we achieve better validation loss using raw text? Is the remark following this figure clarifying that the validation loss has the regularization term only for ULD loss?

Overall, this is a good paper that demonstrates the empirical success of a method designed to solve a previously unconsidered problem.

---

> ### Author Response · Authors · 2024-10-10
>
> We thank reviewer cjLP for their careful review and constructive comments. We are pleased that the reviewer recognizes the importance of the problem and underlines its ease of implementation.
>
> **Regularization and Figure 5:** To leverage all confusion over Figure 5 we bring additional details to the related remark at the top of page 10 with the following paragraph:
> ```
> Even though the cross entropy is lower with raw text data than with ULD Loss, the raw data does not lead to better results. In fact, during training, models distilled with ULD Loss are trained to reproduce the predictions of the teacher model. Consequently, they are not trained to minimize cross-entropy and have maximum confidence in each prediction.
>
> For this reason, a student model trained to reproduce the confidence of a teacher model of 0.92, compared to a model trained only on raw text and trained to predict 1, will have a lower probability on the golden token and consequently a higher Cross Entropy Loss. However, this model will be better calibrated to its true confidence for the sentence.
> ```
>
> Concerning ULD Loss as stabilizer, while the Cross-Entropy is higher for the model distilled with ULD Loss as explained, the range of variation compared to the one trained on raw data remains stable as training progresses, it only increases slightly as the model becomes more aligned with Teacher One and the ULD Loss decreases.
>
> **Complexity in n • log n:** To take into account the sorting step, we updated the complexity value (n to n•log(n)) in Appendix A1.
>
> **Citation:** We modified how citations were written (citep or citet) in the paper to improve reliability and formatting.
>
> **Additional experimentation:** We kindly refer the reviewer cjLP to the comparisons in **Appendix B2**. This ablation performs distillation between a teacher and a student model with the same tokenizer using three different methods: raw text (=0), ULD loss, and KL Div. For clarity, we will bring explicit information that training with (=0) is equivalent to training with raw text in Table 5 description.
>
> We hope we have addressed all the reviewer’s questions and suggestions.

---

> > ### Comment · Reviewer_cjLP · 2024-10-10
> >
> > Appreciate the further clarification: I maintain my original positive review of this paper!

---

### Review · Reviewer_gWp9 · 2024-09-24

**Summary Of Contributions:**

The paper studies the problem of knowledge distillation for LLMs. The paper proposes a novel loss function based on optimal transport, to measure the similarity between the predictive distributions of the teacher and student models. The proposed method is able to handle scenarios where the token spaces of the teacher and student LLMs are different. Extensive ablation studies and experiments demonstrate the strong performance of the proposed distillation method.

**Audience:**

Yes

**Broader Impact Concerns:**

The broader impact is discussed at the end of the paper.

**Claims And Evidence:**

Yes

**Requested Changes:**

Please refer to the points listed under "Weaknesses/Questions" above.

**Strengths And Weaknesses:**

Strengths:
- The paper handles an important problem, which is knowledge distillation for LLMs to which only the output logits are available.
- The proposed loss function is theoretically inspired, because it is based on a notion from optimal transport.
- The performances in the experiments look promising, and indeed verify the consistent advantage of the proposed method over distillation using teacher-generated text.
- The paper is well written. The studied problem if nicely motivated, and the contributions of the paper are clearly discussed.

Weaknesses/Questions:
- I think it would be helpful to add some high-level intuitions about how equation (5) is derived. No need for the technical derivations, some intuitions would suffice.
- Section 3.3, the paragraph "Intuitions": If I understand correctly, the proposed distance in equation (5) does not account for the meaning of the tokens. For example, even if two tokens from the teacher LLM and student LLM represent the same word, this information will not be used by the proposed method. I think this approach may miss important information which is useful for measuring the similarity between the predictive distributions of two LLMs.
- This is related to the point above, at the top of page 6, it is mentioned that "no vocabulary relationship is established", so each transport cost is set to 1. I wonder is it possible to establish relationship between a subset of common vocabularies for the teacher and student? Since I guess there are usually some overlaps between the token spaces of a pair of LLMs, as shown in Fig. 1. This may further improve the performance.
- A limitation of the proposed method is that it requires access to the output logit of the LLM, which makes it not applicable to the currently most powerful LLMs such as ChatGPT and Gemini.
- Figure 4, right plot: why aren't the curves monotonically increasing with the dataset size?

---

> ### Author Response · Authors · 2024-10-10
>
> We thank reviewer gWp9 for their careful review and constructive comments. We are pleased that the reviewer recognizes the problem’s importance and the paper’s contribution.
>
> **About Eq (4):**  It decomposes into two sums:
> - The outer sum iterates over the sequence length (denoted as |x|) to compute the Wasserstein distance at each time step t.
> - The inner sum represents the Wasserstein distance (W1) between the logits of the student and teacher models at that time step.
>
> To compute the W1 distance efficiently, we use a closed-form solution derived under certain assumptions (uniform support length and uniform transport cost, as detailed in **Appendix A**). The inner sum $$\sum_{i=1}^{|\Omega|} \left|\mathbf{p}(x^S_{\sigma^S(i)} | \mathbf{x}^S_{<t}) - \mathbf{q}(x^T_{\sigma^T(i)} | \mathbf{x}^T_{<t})\right|$$ computes the absolute difference between the sorted probability masses of the teacher and student vectors at time step t, ensuring that the overall structure of the teacher's probability distribution is preserved. This process allows the student model to learn not only from the golden tokens (as used in the Cross-Entropy Loss) but also from the surrounding probability distribution, capturing distributional information from the teacher's predictions.
>
> **Further Clarification:** The ULD Loss is specifically designed for black-box scenarios where model architectures and vocabularies vary. This flexibility allows it to be implemented using standard libraries or APIs, while enhancing its accessibility and adoption for practitioners. By not relying on explicit token relationships, ULD Loss avoids vocabulary constraints or complex, manually defined dependencies that could limit its applicability. Concerning improving performances, by relying on explicit relationships. We kindly refer reviewer gWp9 to the experiments detailed in **Appendix B2**. The results show that even when explicit token relationships are maintained across the entire vocabulary, ULD Loss performs on par with  KL Divergence.
>
> **Limitation:** Regarding models that do not provide access to the output logit, we kindly inform that ChatGPT, and more recently Gemini, allow practitioners to access the top K output tokens. Consequently, ULD Loss, compared to classical logit distillation methods requiring a similar vocabulary, enables practitioners to distill these models.
>
> - ChatGPT: https://platform.openai.com/docs/api-reference/chat/create#chat-create-logprobs
> - Gemini: https://github.com/google-gemini/generative-ai-python/issues/238
>
> **About Figure 5**: To ensure robustness, we conducted five additional experiments for each dataset size, constructing confidence intervals for each result. We updated the figure and analysis to reflect these new findings, and we thank the reviewer for prompting this valuable exploration.
>
> We hope we have answered all the reviewer's questions and suggestions.

---

### Review · Reviewer_BZ5v · 2024-09-27

**Summary Of Contributions:**

This paper proposes an effective distillation mechanism that can bridge the gap between teacher and student models with inconsistent architectures and tokenizers. The core of this mechanism is the ULD loss, derived from the optimal transport closed-form solution. The paper's experiments are comprehensive, and the implementation results support the motivation.

**Audience:**

Yes

**Claims And Evidence:**

Yes

**Requested Changes:**

Perhaps some experiments comparing it with logit distillation could be added? As I mentioned above.

**Strengths And Weaknesses:**

**Strengths:**

ULD loss  can bridge the gap between teacher and student models with inconsistent architectures and tokenizers.

**Weakness:**

How would the ULD loss compare to logit distillation in eq (1) if the tokenizers of the teacher and student models were consistent? Can we assume the student model is trained from scratch?

---

> ### Author Response · Authors · 2024-10-10
>
> We thank reviewer BZ5v for their careful review and constructive feedback. We are pleased that the reviewer acknowledges the effectiveness of our distillation mechanism, the comprehensiveness of our experiments, and the support provided by the implementation results.
>
> About the comparison between ULD loss and KLD.  We kindly refer reviewer BZ5v to comparisons in Appendix B2. The results demonstrate that ULD Loss performs on par with the KLD. Ablation on the weighting term lambda is also reported for both losses in Table 5.
>
> **Model Initialization:** In our experiments, all models were initialized using a pre-trained student model due to the computational cost of training from scratch. However, our findings regarding the similar behavior between KL Divergence and ULD Loss suggest that ULD Loss could be equally effective for models trained from scratch.
> One advantage of ULD Loss over KL Divergence is its ability to utilize a smaller vocabulary size, which results in a more compact embedding matrix for the student model compared to the teacher. For instance, using the LLaMA tokenizer vocabulary (128,000 tokens) and a 512-dimensional embedding, the embedding matrix contains approximately 65 million parameters, accounting for 26% of the total model size when including both embedding and decoding layers. We find this direction promising and have included it in our future work considerations.
>
> We hope we have addressed all the reviewer’s questions and suggestions.

---

### Author Response · Authors · 2024-10-10

We thank the reviewers for their careful evaluation and constructive feedback.

We are pleased they recognize the significance of our work, the effectiveness of our method, and the thoroughness of our experiments.

We hope to have addressed all their questions and suggestions and have provided detailed responses to clarify any ambiguities. We believe their comments have strengthened the manuscript (**changes highlighted in blue in the revised PDF**).

---

### Decision · Action_Editor_TJtR · 2024-11-28

**Recommendation:** Accept with minor revision

**Comment:**

This papers studies how to perform distillation with teacher/student models that may be unrelated, e.g., have entirely different model architectures or even tokenizers. This amounts to replacing a standard component in the distillation loss with a version that can accommodate these differences. Specifically, rather than minimizing the KL divergence between the output distributions of the teacher/student models, the authors use a Wasserstein distance term. This is natural but comes with additional complexity, which the authors seek to address through a simple assumption that leads to a clean and efficiently-computed form for the term.

The reviewers are generally positive—this is a reasonable idea with good execution. It is on track for acceptance.

There are a couple of places that the authors could improve the draft:

- Additional experimental evidence. One of the authors’ core ideas is to require that the transport cost is equal to 1, which leads to much higher computational efficiency. The authors note that despite the assumption, this still produces strong results (and verify this experimentally). However, it would be useful to know how much is sacrificed here—how strong would the results have been without this assumption? That would produce a (computationally infeasible) upper bound. If there is indeed a large gap, it would imply that we should continue searching for more efficient ways to do OT without constraining it.

I would suggest that the authors do a simple experiment where they use two synthetic token sets, varying their size (to keep these sufficiently small), run distillation with full OT and their own ULD approach and compare the results.

- As spotted by the reviewers, there are a number of typos, so before camera ready submission, I encourage the authors to do a careful pass.

**Audience:**

Yes, distillation for general pairs of models is of interest to the TMLR audience.

**Claims And Evidence:**

Yes, there is sufficient evidence for the arguments made by the authors.

---

> ### Author Response · Authors · 2024-12-30
>
> We are delighted that the Action Editor and the reviewers have recognized ULD as an innovative idea with solid execution, considering our work to be on track for publication in TMLR.
>
> **Optimal transport with non-uniform costs:** To evaluate the impact of the uniform transport cost assumption on the ULD loss, we conducted a new set of experiments comparing the results of distillation with a uniform cost and two non-uniform cost matrices (grounded on embedding distance & Levenshtein distance). These experiments, along with the detailed experimental setup, are presented in Appendix F, page 28.
>
> For all experiments in our limited setting, we do not observe consistent improvements in model performance when trained with ULD Loss and uniform transport costs. However, training with ULD Loss using a uniform cost significantly accelerates distillation compared to non-uniform costs (up to 4.49× faster). This highlights the practical advantage of the uniform cost assumption in achieving computational efficiency.
>
> We thank the Action Editor for this experimental suggestion, which we believe adds substantial value to the paper. We hope future research will explore alternative cost matrices while maintaining stable computational complexity.
>
> With regard to the typos, we have taken particular care in re-reading the paper and hope to have corrected them all.
>
> We thank the reviewers and Action Editor for their thorough evaluation and constructive feedback, which have significantly improved the readability of our paper and the robustness of our evaluation experiments. We hope we have addressed all their questions and suggestions.

---

> > ### Comment · Action_Editor_TJtR · 2025-01-08
> > **Missing Author Names in Camera Ready**
> >
> > Hi Folks,
> >
> > The content looks good --- can you all add your names to the camera ready pdf?
> >
> > Best,
> >
> > Your AC

---

> > > ### Author Response · Authors · 2025-01-09
> > > **Authors' names added.**
> > >
> > > Hello, and Happy New Year to the reviewers and AC!
> > >
> > > We have just added the authors’ names to the paper.
> > > I would like to sincerely thank the reviewers and AC one last time for their excellent work and constructive feedback, which made the review process nice and enriching.